# Transcriptional Characteristics Showed That miR-144-y/FOXO3 Participates in Embryonic Skin and Feather Follicle Development in Zhedong White Goose

**DOI:** 10.3390/ani12162099

**Published:** 2022-08-17

**Authors:** Ichraf Mabrouk, Yuxuan Zhou, Sihui Wang, Yupu Song, Xianou Fu, Xiaohui Xu, Tuoya Liu, Yudong Wang, Ziqiang Feng, Jinhong Fu, Jingyun Ma, Fangming Zhuang, Heng Cao, Honglei Jin, Jingbo Wang, Yongfeng Sun

**Affiliations:** 1Department of Animal Genetics, Breeding and Reproduction, College of Animal Science and Technology, Jilin Agricultural University, Changchun 130118, China; 2Key Laboratory of Animal Production, Product Quality and Security, Jilin Agricultural University, Ministry of Education, Changchun 130118, China

**Keywords:** skin, feather follicles development, goose, transcriptome sequencing, gene expression

## Abstract

**Simple Summary:**

Feather is one of the most valuable and economical products in goose farming and plays a crucial physiological role in birds. For avian biology and the poultry industry, it is essential to comprehend and regulate how skin and feather follicles develop during embryogenesis. This study showed that several key regulatory genes (FOXO3, CTGF, and PTCH1, among others) and miRNAs (miR-144-y) participated in the developmental process of the skin and feather follicles in Zhedong white goose. Our findings are particularly important because they will serve as a valuable resource for upcoming studies on down feathers in agricultural economic growth regarding complex molecular mechanisms and breeding techniques.

**Abstract:**

Skin and feather follicle development are essential processes for goose embryonic growth. Transcriptome and next-generation sequencing (NGS) network analyses were performed to improve the genome of Zhedong White goose and discover the critical genes, miRNAs, and pathways involved in goose skin and feather follicle morphogenesis. Sequencing output generated 6,002,591,668 to 8,675,720,319 clean reads from fifteen libraries. There were 1234, 3024, 4416, and 5326 different genes showing differential expression in four stages, E10 vs. E13, E10 vs. E18, E10 vs. E23, and E10 vs. E28, respectively. The differentially expressed genes (DEGs) were found to be implicated in multiple biological processes and pathways associated with feather growth and development, such as the Wnt signaling pathway, cell adhesion molecules, ECM–receptor interaction signaling pathways, and cell cycle and DNA replication pathways, according to functional analysis. In total, 8276 DEGs were assembled into twenty gene profiles with diverse expression patterns. The reliability of transcriptome results was verified by real-time quantitative PCR by selecting seven DEGs and five miRNAs. The localization of forkhead box O3 (FOXO3), connective tissue growth factor (CTGF), protein parched homolog1 (PTCH1), and miR-144-y by in situ hybridization showed spatial-temporal expression patterns and that FOXO3 and miR-144-y have an antagonistic targeting relationship. The correlation coefficient of FOXO3 and miR-144-y was -0.948, showing a strong negative correlation. Dual-luciferase reporter assay results demonstrated that miR-144-y could bind to the expected location to suppress the expression of FOXO3, which supports that there is a targeting relationship between them. The detections in this report will provide critical insight into the complex molecular mechanisms and breeding practices underlying the developmental characteristics of skin and feather follicles in Zhedong white geese.

## 1. Introduction

Avian feathers, similar to mammalian hairs and reptile scales, are skin appendages [1,2]. Feathers can be broadly divided into two broad categories, orthodermal feathers, which are produced by primary feather follicles, and down feathers, which are produced by secondary feather follicles [3] and gradually emerge during the embryonic stage of the bird’s life along with the development of the skin and mature after birth. Poultry down feathers, especially waterfowl, have high economic value and are highly valued in the light industry for their warmth and hydrophobicity. In addition to being valued by animal breeders for their relevance to down production, poultry feather follicles have also been studied by biologists in recent years as an excellent model for regeneration and development [4].

Common poultry such as chickens, ducks, and geese are prematurely feathered birds, as evidenced by the fact that the chicks are covered in feathers after birth, indicating that within just three to four weeks of incubation, feather follicles have already differentiated and developed from primitive stem cells and developed fetal feathers, a process that is orderly and dramatic. However, the process of feather follicle development from nothing to something can be broadly divided into four processes: The signal response phase, microstructure formation, macrostructure formation, and morphological structure formation [5,6]. During this process, a histological perspective can be observed in the gradual formation of smooth skin into dermal condensates, followed by feather bud formation and finally the formation of a complete feather follicle structure. It has been widely reported that several signals are responsible for the regulation of this orderly process, including transforming growth factor-beta (TGFβ) signaling [7], bone morphogenetic protein (BMP) signaling [8,9,10], sonic hedgehog (Shh) signaling [11,12], NOTCH signaling [13], and wingless-related (Wnt) signaling [3,14], which is the core signal for feather follicle formation. These signals, together with non-coding RNAs, form a complex and orderly regulatory network under which a miracle of life is accomplished in a matter of weeks. With the progressive understanding of the developmental process of feather follicles, more and more new key regulatory factors are being uncovered in addition to the signals mentioned above. The advent of next-generation sequencing (NGS) has become a powerful tool to uncover the molecular signaling involved in the formation of feather follicles for feather maturation [15,16,17,18,19,20]. While the feather follicles of chickens and ducks are increasingly well studied, geese, as poultry with extremely high down feather quality, have been relatively seldomly studied.

In this study, we used the native Chinese Zhedong white goose, a medium-sized and higher-quality down-producing goose breed, as the subject. High-throughput technology and deep sequencing analysis were employed to screen differentially expressed mRNA and microRNAs (miRNAs) among 15 comparison groups from five important embryonic developmental stages (E10, E13, E18, E23, and E28). Finally, we believe that the results of this study will attract the attention of research groups and could be valuable for molecular research on the basic mechanism underlying feather formation.

## 2. Materials and Methods

### 2.1. Ethics Statement

In this experiment, all procedures for animal experiments were approved by the Animal Health Care Committee of Animal Science and Technology College of Jilin Agricultural University (Approval No. GR (J) 18-003).

### 2.2. Sample Collection

In total, 200 fertilized eggs of Zhedong white geese were hatched in an auto-incubator, and samples were collected on the 10th (E10), 13th (E13), 18th (E18), 23rd (E23), and 28th (E28) days of the embryonic stage, respectively, with 40 eggs collected at each time point, and the geese gender was detected using blood direct amplification PCR. All the samples tested in this study were males. Nine skin samples containing follicles were collected from the dorsum of goose embryos and used to mix samples for RNA-seq and small RNA-seq at each time point, and every three samples were mixed into one. Among them, three samples at each stage were used for RNA extraction and three samples were set with 4% paraformaldehyde (DEPC treatment), dried out, and then embedded after 48 h with a common method for histological staining and in situ hybridization assay.

### 2.3. Hematoxylin and Eosin Staining

The sections were sequentially placed in xylene for 2 periods of 10 min each for rehydration with different concentrations of ethanol (95%, 90%, 80%, and 70%) for 5 min and then rinsed with distilled water. Thus, the cut slices were stained with Harris hematoxylin solution for 3–8 min, following treatment with 1% hydrochloric acid-ethanol, and then the slides were washed with tap water. The sections were then placed in 1% ammonia to bring them back to blue and rinsed again with water. Subsequently, to remove the excess dye and for onward dehydration, an eosin solution and a graded series of ethanol were used. Thereafter, each slide was sealed with neutral resin after being exposed to xylene for 10 min. Finally, under the Nicon-300 light microscope, the morphological changes in feather follicles were detected (Nicon, Tokyo, Japan).

### 2.4. Masson Staining

Slides were stained in Weigert’s iron hematoxylin solution for 5–10 min and then rinsed with water, following treatment with 1% hydrochloric acid alcohol differentiation and then washed again with trap water for several minutes. Subsequently, the slides were dyed with Biebrich scarlet acid fuchsin solution for 5–10 min, rinsed with distilled water, incubated in 1% phosphomolybdic acid aqueous solution for 5 min, dyed with aniline blue solution for 5 min, and fixed in 1% glacial acetic acid for 1 min. Consequently, a graded series of ethanol was used to wash out the excess dye and for onward dehydration. Finally, each slide was treated with xylene for 10 min and sealed with neutral resin.

### 2.5. Van Gieson Staining

Paraffin blocks were deparaffinized and hydrated with distilled water. Subsequently, the sections were stained for 5 min with Weigert’s iron hematoxylin solution and washed with tap water for 5–10 min. After that, the slides were dyed with Van Gieson’s solution for 1–5 min, dehydrated in 95% alcohol rapidly for several seconds, and the dehydration was completed in three changes of absolute alcohol. Thereafter, each slide was sealed with neutral resin after being cleaned with xylene.

### 2.6. RNA Isolation, Library Creation and Sequencing

Following the manufacturer’s instructions, total RNA was extracted using the TRIzol Reagent kit (Invitrogen Life Technologies, Carlsbad, CA, USA). To remove DNA contamination from RNA samples, DNase I (Ambion, Austin, TX, USA) was used. Using RNase-free agarose gel electrophoresis and the Agilent 2100 Bioanalyzer (Agilent Technologies, Palo Alto, CA, USA), respectively, the quantity and integrity of total RNA were evaluated. Oligo (dT) beads were used to enrich eukaryotic mRNA after total RNA had been extracted, while prokaryotic mRNA was enriched by removing rRNA with the Ribo-ZeroTM Magnetic Kit (Epicentre). Using a fragmentation buffer, the enriched mRNA was fragmented into small fragments and the NEBNext Ultra RNA Library Prep Kit for Illumina (NEB #7530, New England Biolabs, Ipswich, MA, USA) was used to reverse-transcribe it into cDNA. The cDNA fragments were purified using the QiaQuick PCR extraction kit (Qiagen, Hilden, Germany) and then end-repaired, poly (A) was added, and the fragments were ligated to Illumina sequencing adapters after second-strand cDNA synthesis with DNA polymerase I, RNase H, dNTP, and the buffer. The ligation reaction was purified using AMPure XP Beads (1.0×), and the size of the ligation products was determined using polymerase chain reaction (PCR) amplification and agarose gel electrophoresis. The generated cDNA library was sequenced by Gene Denovo Biotechnology Co. (Guangzhou, China) using Illumina Novaseq6000.

### 2.7. MicroRNA Library Creation and Sequencing

The RNA molecules in the size range of 18–30 nt were enriched by polyacrylamide gel electrophoresis (PAGE) after the total RNA was isolated using the TRIzol Reagent kit (Invitrogen Life Technologies, Carlsbad, CA, USA). Then, the 36–44 nt RNAs were enriched after the 3′ adapters were inserted and the RNAs were also ligated to the 5′ adapters. PCR amplification was used to reverse transcribe the ligation products and the 140–160 bp sized PCR products were then purified to create a cDNA library and sequenced using Illumina HiSeqTM 2500 by Gene Denovo Biotechnology Co. (Guangzhou, China).

### 2.8. PCA, Quantification, and Analysis of DEGs and Clustering Analysis

The R package gmodels. Available online: http://www.r-project.org/ (accessed on 11 August 2021) was used to perform principal component analysis (PCA). The StringTie v1.3.1 software was used to compile each sample’s mapped reads into a reference database. For each transcription region, a fragment per kilobase of transcript per million mapped reads (FPKM) value was calculated to quantify its expression abundance and variations using RSEM software. The FPKM approach may directly compare the variation in gene expression between samples by removing the effects of various gene lengths and sequencing data amounts. The DESeq2 software between two separate groups and the edgeR package. Available online: http://www.r-project.org/ (accessed on 11 August 2021) between two samples were used to identify the DEGs. However, the false discovery rate (FDR) < 0.05 and a |Fold Change| ≥ 2 were assembled to enrich the significant differentially expressed transcripts or genes. After that, the DEGs were used for GO functions and KEGG pathway enrichment analysis. Significantly enriched terms and pathways were those GO terms and KEGG pathways with *p* < 0.05. The DEGs were grouped using the Short Time-series Expression Miner software (STEM) to evaluate the expression pattern of all differential genes and then the variables were selected (-pro 20 -ratio 1.0000 [log2 (2) = 1, log2 (1.5) = 0.5849625, log2 (1.2) = 0.2630344]) for trend analysis. Next, GO/KEGG functional enrichment analyses were performed for the genes in each trend, and the *p*-value was calculated by hypothesis testing. The *p*-value obtained was corrected by FDR, with a *p*-value ≤ 0.05 as the minimum, and the GO term and pathway that were significantly enriched in that trend satisfied this condition.

### 2.9. GO Enrichment Analysis and Pathway Enrichment Analysis

For the identified genes relevant to the development of skin and feather follicles, GO function analysis and KEGG pathway analyses were performed. Gene ontology (GO) enrichment analysis is a systematic approach for annotating gene functions. In the Gene Ontology database. Available online: http://www.geneontology.org/ (accessed on 7 August 2021), all DEGs were first mapped to GO terms, the numbers of genes for each term were obtained, and the hypergeometric test was used to identify GO terms that were significantly enriched in DEGs as compared to the genome background. The calculated *p*-value went through FDR correction, taking FDR ≤ 0.05 as a threshold. GO terms meeting this condition were defined as significantly enriched GO terms in DEGs using the following calculation:$$P=1-\overset{m-1}{\underset{i=0}{\sum }}\frac{\left(\begin{array}{c} M\\ i \end{array}\right)\left(\begin{array}{c} N-M\\ n-i \end{array}\right)}{\left(\begin{array}{c} N\\ n \end{array}\right)}$$

In which *N* represents the total number of genes with GO annotation, *n* is the DEGs number in *N*, *M* represents the total number of genes annotated to a specific GO term, and *m* is the DEGs number in *M*. This approach can identify major biological functions that DEGs perform.

Genes usually interact with each other to play roles in certain biological functions. Pathway-based analysis can be useful to better understand the gene’s biological functions. The main public pathway-related database is the Kyoto Encyclopedia of Genes and Genomes (KEGG). Available online: http://www.genome.jp/kegg (accessed on 7 August 2021). To further identify the significantly enriched metabolic pathways or signal transduction pathways in DEGs compared with the whole genome background, pathway enrichment analysis should be conducted. Calculations were performed using the same formula that was applied to the GO analysis.

### 2.10. MicroRNA Target Genes Prediction

MicroRNA targets were predicted using three software, RNAhybrid (Rehmsmeier, Marc and Steffen, Peter and Hoechsmann, Matthias and Giegerich, Robert Fast and effective prediction of microRNA/target duplexes RNA, RNA, 2004. Version 2.1.2) + svm_light (Thorsten Joachims, Developed at University of Dortmund, Informatik, AI-Unit Collaborative Research Center on ‘Complexity Reduction in Multivariate Data’ (SFB475). Version 6.02), Miranda (Version 3.3a), and TargetScan (Version 7.0). The TargetScan website. Available online: http://www.targetscan.org (accessed on 1 March 2022) provided the microRNA sequences and family information.

### 2.11. MicroRNA Target Network Construction

The Pearson correlation coefficient (PCC) was used to analyze the expression correlation between miRNA and target. All RNAs were expressed differently in the multi-group, and pairs with PCC < −0.7 and *p* < 0.05 were chosen negatively as co-expressed miRNA-target pairs.

### 2.12. RNA Isolation

Tissues were ground in Eppendorf tubes using a hand-held homogenizer, and 1 mL of RNAiso Plus reagent was added to each tube and left to stand at room temperature. The supernatant was taken after centrifugation, and chloroform was added for phase separation following the manufacturer’s instructions. The aqueous phase was taken for isopropanol precipitation, 75% ethanol (DPEC water preparation) was used to wash the precipitate, and finally, the RNA was dissolved with DEPC water.

### 2.13. Quantitative Real-Time PCR (RT-qPCR) Validation of Sequencing Data

In total, eight DEGs (forkhead box O3, FOXO3; connective tissue growth factor, CTGF; protein parched homolog1, PTCH1; N-Myc down-regulated gene1, NDRG1; fibroblast growth factor binding protein 1, FGFBP1; transcriptional repressor GATA Binding 1, TPRS1; microfibril associated protein 5, MFAP5) and five miRNAs (let-7-y, miR-103-y, miR-107-z, miR-181-y, miR-183-x) related to feather development were selected and validated the accuracy of sequencing data by RT-qPCR. First, the first-strand cDNA synthesis was conducted following the protocol of the reverse transcription kit (Thermo, Waltham, MA, USA), including both genomic DNA removal and reverse transcription reactions. Using the SYBR Green technique (Monad Biotech Co., Ltd., Wuhan, China), the expression of associated genes was found. 18S rRNA and GAPDH were selected as double internal references for the assay, and the quantification of genes was conducted using the 2^−ΔΔCt^ method. The reverse transcription of miRNA was performed by the stem-loop method (Vazyme, Nanjing, China), and the reverse transcription reaction was completed using specific reverse transcription primers following the protocol. Using both specific and universal primers, the fluorescence quantitative PCR assay was performed with snRNA U6 as the internal control gene. All the used primers are listed in Table 1.

### 2.14. Dual-Luciferase Reporter Gene Assay

The design and synthesis of the dual-luciferase reporter gene vectors were carried out by Shanghai Biotech Biotechnology Co. (Shanghai, China). Two vectors, FOXO3-MU and FOXO3-WT, were designed to represent the 3′UTR mutation and wild-type of the FOXO3 gene, respectively. The vectors and the miRNA mimics were co-transfected with HEK 293T cells prepared in advance for transfection into 96-well plates, and four assay groups were obtained by combining NC mimics, miR-144-y, FOXO3-MU, and FOXO3-WT vectors, with three biological replicates in each group. The Promega Dual-Luciferase system was employed to carry out the assay. However, 5× passive lysis buffer (PLB) was diluted to 1× PLB with distilled water, added at 100 μL per well of a 96-well plate and, the cells were dispersed by pipetting with a pipette gun and placed on a shaker for 15 min at room temperature. The cell lysate was aspirated into a 1.5 mL centrifuge tube and centrifuged at 4 °C for 10 min at 12,000 rpm. After 15 min of slow shaking, the supernatant was then collected and placed in a second tube. Then, 100 μL of the Luciferase Assay Reagent II (LAR II) (Luciferase Assay Reagent, Progema) working solution was added to the 96-well plate, 20 μL of cell lysate was added, and the plate was pipetted. The Firefly luciferase value, which is the internal reference value, was measured and recorded after the plate had been mixed 2–3 times. Thereafter, the Renilla luciferase value was measured, which is the reporter gene luminescence value after adding 100 μL of the Stop & Glo^®^ Reagent (Luciferase Assay Reagent, Progema, Madison, WI, USA) and pipetting the plate 2–3 times.

### 2.15. In Situ Hybridization

Slides (5–10 μm thickness) containing dorsal skin tissue sections embedded in paraffin were baked at 62 °C for 2 h, dewaxed in xylene twice for 15 min each, hydrated in successive grades of alcohol, and then washed with DEPC water. The cut pieces were simmered for 10–15 min in the repair solution and cooled naturally. The slides were blow-dried, then 20 g/mL proteinase K was added, and they were incubated for 20–30 min at 37 °C before being carefully washed with pure water. These slides were additionally cleaned with PBS solution 3 times for 5 min each, and then 3% methanol-H_2_O_2_ was added dropwise. Subsequently, the sections were placed in PBS (pH 7.4) on a decolorization shaker and incubated at room temperature in the dark for 15 min before being rinsed three times for 5 min each. A prehybridization solution was dropwise added to the slides, and they were then incubated at 37 °C for 1 h. Slices were hybridized overnight at 37 °C at a constant room temperature after the prehybridization solution had been decanted and the probe hybridization solution had been added dropwise. The probe was designed and synthesized by Wuhan Servicebio Technology Co., Ltd. (Wuhan, China). The specific probe sequence was designed by the solid-state phosphite Triester method, which was subjected to a four-step cycle of deprotection, coupling, capping, and oxidation. One nucleotide monomer at a time was attached to the 3′ solid-phase carrier, and the modifications are labeled at the corresponding positions in the process. The synthesized oligonucleotides were chemically cut off from the solid carrier (CPG) and the protective group was removed by ammonia dissociation. After the crude product passed the quality test, the crude product was purified by HPLC, and the target product was prepared by separating impurities with different degrees of separation. The pure product was detected by the LTQ-MS mass spectrometer. The qualified pure product was quantificationally packed with A260 by an enzyme label analyzer, and then the sample was dried to a dry powder by a vacuum dryer (Table 2). After incubation, the hybridization solution was removed using 2 × SSC, 1 × SSC and 0.5 × SSC at 37 °C for 10 min each time. For 10 min at room temperature, a drop of blocking serum BSA was added and then it was poured off. After that, the slices were added with a drop of anti-DIG-HRP (Jackson, 200-032-156) and incubated at 37 °C for 40 min and washed with PBS for 20 min. Subsequently, the sections were re-stained for 3 min with Harris hematoxylin solution and then were washed with tap water, following the treatment with 1% hydrochloric acid alcohol for a few seconds and the slides were washed again with distilled water. A gradient series of alcohol was then used to wash off the excess dye and for further dehydration after the dye had been turned to blue using ammonia and rinsed with running water. Afterward, each slide had a xylene treatment before being slightly dried and sealed with neutral resin.

### 2.16. Statistical Analysis

Statistical analyses were conducted using R software (R Core Team (2020). R: A language and environment for statistical computing. R Foundation for Statistical Computing, Vienna, Austria. Version 4.1.3) and Excel software (Microsoft Corporation One Microsoft Way Redmond, WA 98052-7329, USA. Version 2013). GraphPad Prism 8 was used to visualize the data (GraphPad, San Diego, CA, USA). The Pearson correlation between the results of RNA-seq and qPCR was analyzed using data correlation analysis. A two-tailed Student’s t-test was used in the comparison of dual-luciferase reporter gene assay. The significant difference in the results was deemed as *p* < 0.05. In the figures, the results are presented as mean± Standard Error of the Mean.

## 3. Results

### 3.1. Histological Structure of Skin and Feather Follicles at Different Stages

To visualize the developmental pattern of the embryonic skin and feather follicles of Zhedong white goose at five embryonic developmental stages (E10, E13, E18, E23, and E28), we collected the results of its dorsal appearance with three different staining methods (Figure 1). The skin of the goose embryo was smooth and transparent on the dorsal surface at the early embryonic stage of E10. By E13, punctate plumage structures appeared on the dorsal surface, and by E18, the dorsal tract was almost entirely covered with feathers. With these changes at the last three time points, the embryo became larger, and the plumage color gradually changed from white to yellow. The results demonstrate that no major feather follicle structures have yet developed at the early embryonic stage of E10. Short feather buds were discernible in the feather tracts and a single layer of skin had developed by E13; however, these structures were the locations of the future feather follicles. Skin thickness increased at E18 (primitive period of secondary feather follicles), and the typical primary feather follicle structures appeared and were accompanied by the beginning of the development of smaller secondary feather follicles. The latter two stages (E23 and E28) mainly show the process of change in the size of primary feather follicles, the development of secondary feather follicles, and the increases in skin thickness on a temporal gradient. With the help of Masson staining and VG staining, the differentiation of collagen fibers in the skin was observed (Figure 1). The results indicated that the differentiation of the dermis from the epidermis likely started between E13 and E18 and gradually differentiated and matured over subsequent time. Furthermore, it has been widely noticed that the differentiation of the dermis from the epidermis was accompanied by the development of feather follicles in parallel.

### 3.2. Overview of RNA Sequencing Data

In order to verify the DEGs and establish the transcriptome reference, a total of fifteen RNA-seq libraries were sequenced from five groups of goose embryonic skin tissue at five development stages (E10, E13, E18, E23, and E28) using three biological replicas, respectively. These stages were chosen to cover the whole initiation and development process of feather follicles in each embryonic stage. After eliminating the poor-quality reads, clean reads were obtained with a yield from 6,002,591,668 to 8,675,720,319 with an average of 97.70% Q20 base and 93.79% Q30 base for each sample. Approximately 84.54% to 85.95% of the reads can be mapped to the reference Zhedong white goose (*Anser cygnoides domesticus*) genome (GCF_000971095.1). We further analyzed the relationship between the samples using Principal Component Analysis (PCA) and selected the top two principal components to plot the results (Figure 2). PCA results indicate that for PC1, the samples of E10 and E13 are relatively close to each other, while the remaining three groups are clearly separated from these two groups. For PC2, the samples of E10 and E13 can be separated relative to PC1, while the remaining three groups can still be obtained with good separation. The clustering within groups was excellent for both PC1 and PC2, indicating that the subsequent differences obtained between groups could be a relatively realistic reflection of the biological differences between samples.

### 3.3. GO Enrichment and KEGG Pathway Analysis of the Differentially Expressed Genes

To explore potential genes regulating feather follicle development, we compared samples of E13, E18, E23, and E28 with E10, respectively, screened the genes with FDR < 0.05 and |log2 FC| > 1 as significantly differentially expressed genes, and obtained four sets of different genes, 1234 (E10 vs. E13, 624 up-regulated, 610 down-regulated), 3024 (E10 vs. E18, 1663 up-regulated, 1361 down-regulated), 4416 (E10 vs. E23, 2103 up-regulated, 2313 down-regulated), and 5326 (E10 vs. E28, 2283 up-regulated, 3043 down-regulated). All subsequent analyses were completed based on these differential gene clusters. GO and KEGG enrichment analyses were carried out to further discover the functions of these DEGs. In the GO analysis, we focused on the top three secondary GO terms for differential genes (Figure 3). The DEGs were classified into three main GO categories based on the GO annotation: Biological process (cellular process, single-organism process), cellular component (cell part, cell, and organelle), and molecular function (binding, catalytic activity, molecular transducer activity). Besides these consistent enrichment results, we also discovered that some of the genes differing between E13 and E18 and E10 were enriched for biological regulation entries, while the other two groups (E23 and E28) were enriched for metabolic processes, and these commonalities and differences were the focus of our study.

The KEGG annotation helps us to better understand the changes in signaling pathways (Figure 4). We focused on the top twenty differential pathways enriched for each group of differential genes and, as with the GO analysis, we identified some pathways that were present in all groups of results, namely the neuroactive ligand–receptor interaction and the ECM–receptor interaction, while cell adhesion molecules (CAMs) were present in the rest of the results except for the E10 vs. E28 comparison group. Notably, signaling pathways regulating the pluripotency of stem cells suggest that some stem cell-related activity occurred in the dorsal skin at both times. Interestingly, some genes in the cell cycle and DNA replication pathways were altered in the skin that developed in late embryonic stages (E23 and E28) compared to E10, which may be related to greater changes in the biological properties of late embryonic skin as it matures. Finally, we found that some of the differential genes in E10 vs. E28 were enriched in the histidine metabolism, glycerolipid metabolism, and Wnt pathways, which are known to be the initiating signaling pathways for feather follicle development, while changes in the histidine and glycerolipid metabolism pathways may be reflected in the progressive development and activation of these pathways as the skin follicles develop.

### 3.4. Trend Analysis of DEGs among the Developmental Stages

Expression profiles can reveal the various and intricately complex interactions of genes. Genes with identical functions might have related expression patterns. We performed a clustering analysis of all 8276 DEGs with a time gradient to further characterize the significant expression profiles related to the development of feather follicles and obtained a total of twenty gene profiles, which were numbered from zero to nineteen (Figure 5A,B), in which 5989 were clustered into four profiles: Profile 0 (2869 genes), profile 16 (692 genes), profile 18 (616 genes), and profile 19 (1812 genes). These four profiles represent four different trends, namely the pre-response (profile 0), which gradually down-regulated, the mid-response, which first up-regulated and then leveled off before finally declining and includes the pre-mid response (profile 16) and the mid-late response (profile 18), and, finally, the late response, which gradually up-regulated (profile 19) (Figure 5C,F,I,L).

As with the differential gene analysis, we continued to focus on the top three enriched genes in the level2 GO terms, and the GO analysis showed high consistency in the secondary GO terms (Figure 5D,G,J,M), with most genes enriched in the biological process (cellular process, single-organism process), cellular component (cell part, cell, organelle), and molecular function (binding, catalytic activity, molecular function), while profile 0 and profile 16 had some genes enriched in the metabolic process term and profiles 18 and 19 had some genes enriched in biological regulation. However, this result is in accordance with the findings of our differential gene analysis, and the functional changes resulting from the difference between the pre-mid response genes and the mid-late response genes may be concentrated in the two main categories of metabolic process and biological regulation.

Interestingly, among the 20 pathways analyzed by KEGG, such differences were more pronounced in the pre-mid and mid-late phases (Figure 5E,H,K,N). Profile 0 pathways were focused on DNA and RNA changes and cell proliferation and division functions, such as the cell cycle, DNA replication, RNA transport, and homologous profiles. In addition, profile 16 and profile 18 were more enriched in functional pathways such as the PPAR signaling pathway, sphingolipid, cell adhesion, and the ECM–receptor interaction. In the late-response profile 19, the pathways of interest were the neuroactive ligand–receptor interaction, the ECM–receptor interaction, the cell adhesion signaling pathway, the receptor interaction, and cell adhesion molecules.

### 3.5. Validation of RNA and miRNA Expression Results by qRT-PCR

To confirm the validity of the RNA-Seq and miRNA-seq results, we chose five miRNAs and seven highly expressed mRNAs associated with feather formation to conduct the RT-qPCR test at five time points: E10, E13, E18, E23, and E28, and correlated the results with the sequencing results. As shown in Figure 6, these qPCR results were highly correlated with the sequencing results, and most of the correlation coefficients were within the ideal range (R2 > 0.8), which proved the reliability of our RNA-seq and miRNA-seq results.

### 3.6. In Situ Hybridization Assay

The in situ hybridization assay was used to explore DEG expression and miR-144-y expression and localization at different developmental stages of goose embryos. As shown in Figure 7, CTGF was expressed in the skin epidermis at E10 and on the surface of the skin and feather bud at E13. The expression of CTGF was found on the skin and feather follicle but less on the skin at E18. However, CTGF was mainly concentrated in the root of the feather follicle while it was less in the skin at E23, and it was also more expressed in the feather follicle at E28. Meanwhile, PTCH1 was expressed in the skin epidermis at E10 and was found on the surface of the skin and bulge feather bud, but less in the skin at E13. The expression of PTCH1 was obvious in the skin epidermis and feather follicle, but less in the skin at E18 and primarily concentrated in the root of the feather follicle at E23, as well as PTCH1 was found in the feather follicle at E28. The FOXO3 gene was not expressed on the surface of the skin at E10, while miR-144-y was found in the skin epidermis at E10. The expression of FOXO3 was only obvious in the feather follicle and it was not found in the bulged feather bud at E13. However, the expression of miR-144-y was found on the surface of the skin and in the bulged feather bud at E13. Moreover, FOXO3 was expressed in the feather follicle and less expressed in the skin at E18. Compared with FOXO3, miR-144-y was expressed in the skin epidermis and feather follicle, but less in the skin at E18. At E23, FOXO3 was more expressed in the skin and almost not in feather follicle, while mir-144-y was expressed in feather follicle and mostly concentrated in the root of feather follicle even if it was less in the skin at E23. FOXO3 was rarely found in the feather follicle at E28. Furthermore, miR-144-y was more expressed in follicles than FOXO3 at E28. These results suggest that FOXO3 and miR-144-y have an antagonistic targeting relationship.

### 3.7. Construction Analysis of miRNA and mRNA Network

To investigate the potential regulatory network during feather follicle development in the embryonic skin of the Zhedong white goose, we mapped the miRNA-mRNA association network based on correlation (Figure 8A). MiRNAs primarily play a role in biological regulation through interactions with their target mRNAs, which frequently results in downregulated mRNA expression. Here we focused on some genes that may be associated with feather follicle development and skin differentiation, including CTGF, FOXO3, WNT Inhibitory Factor 1 (WIF1), Frizzled Class Receptor 6 (FZD6), and PTCH1, with highly probable miRNAs also involved in feather follicle development and skin differentiation, including miR-92-y, miR-182-x, miR-144-y, miR-128-y, and other miRNAs and they were negatively associated with the above mRNAs. In our network diagram, we noted a high number of genes and small RNAs pointing to the transcription factor FOXO3. There have been findings on the role of FOX family genes in the development of hair/feather follicles. We, therefore, hypothesize that FOXO3 may be crucial to this network of interactions, and among the miRNAs targeting FOXO3, miR-144-y correlated more with FOXO3 (corr = −0.948, *p* < 0.05). So, to further confirm the targeting relationship, we used the dual-luciferase reporter gene assay (Figure 8B). The findings demonstrated that miR-144-y significantly down-regulated luciferase expression of FOXO3-3′UTR-WT compared to the NC group (*p* < 0.01), demonstrating a binding interaction between the two in this assay, and the targeting relationship was established. After the mutation, the expression of the FOXO3-3′UTR-MUT luciferase was not down-regulated by miR-144-y compared to the NC group (*p* > 0.05), demonstrating that the mutation was successful (Figure 8C).

## 4. Discussion

Feathers are an important poultry product with high economic value and serve numerous physiological functions in birds, such as thermoregulation (downy feathers), physical protection, and tactile ability [21]. During embryogenesis, feather follicle development, and especially the secondary feather follicle, is one of the critical bioprocesses that determine the market value of poultry down feathers [22]. The Zhedong white goose, which is characterized by high-quality down feather production, makes it a great model for functional genomic studies of poultry. Prior research has shown that between the ages of E12 and E14, goose skin begins to develop short feather buds [3,15,23], but in our study, histomorphological observations of the Zhedong white goose embryonic skin revealed that the formation of short feather buds appeared at E13, proposing that feather bud initiation may be linked to variations in goose breeds or hatching conditions. Additionally, the outcomes of this study demonstrated that feather follicle morphogenesis was finished by E18, which was congruent with earlier findings in chickens at E15 [24]. Thus, it was hypothesized that feather follicles form earlier in the chick embryo than in the goose, presumably due to the longer embryonic development period in geese (31 d) compared to chickens (21 d). Furthermore, studies have reported that secondary feather follicles develop most fully at 28 days of embryonic age, and at this stage, the dorsal track of embryos is totally covered by feathers [3,15,23], which was clearly observed in our results.

Recently, using high-throughput technologies, several studies have focused on the molecular and regulatory mechanisms involved in skin and feather/hair follicle development [13,14,15,17,19,25,26,27,28,29]. In the present study, we performed next-generation sequencing (NGS) to discover the intricate molecular processes that underlie the growth of skin and feather follicles in the Zhedong white goose as well as any potential new regulatory factors that might be involved and the transcript profiles were identified at five different development stages (E10, E13, E18, E23, and E28).

The development of the epidermis and different skin appendages, including feather/hair follicles, occurs as a result of the complicated and dynamic process of skin and feather growth [30]. However, the growth of feather follicles is influenced by changes in a variety of cellular activities in the follicle, which are in turn controlled by several transcription factors, signaling pathways, and epigenetic control. Here, the GO analysis helped to clarify the primary roles of the DEGs, and the KEGG pathway analysis helped to pinpoint the principal pathways in which the DEGs are active. The upregulated DEGs in this profile were abundant in cell parts, cellular processes, catalytic activity, single-organism processes, molecular transducer activity, and binding, and some of them at E10 vs. E13 and E10 vs. E18 were enriched in biological regulation and metabolic processes, which were closely associated with feather follicle development. Similarly, numerous studies have also shown that the results of GO analysis of the DEGs were mostly enriched in those pathways, which are likely related to the development of the skin and feather follicles in *Anser cygnoides* and *Anser Anser* [15,17]. KEGG, as a collective database, is used for a systematic analysis of gene functions [29]. Cell adhesion molecules (CAMs), the neuroactive ligand–receptor interaction pathway, the ECM–receptor interaction, and the Wnt signaling pathway were the main pathways anticipated in the current study and were in line with the findings in the embryonic skin of chickens and geese [17,19,31]. Most of these signaling pathways are known to control a variety of biological cell functions, including cell proliferation and survival, morphogenesis, the maintenance of structure and function [32], and skin and hair/feather follicle development in humans [33], chickens [6], cashmere goats [34,35], and mice [36]. Along with the typically known signaling pathways in each of the five stages, some additional signaling pathways were enriched according to the developmental stage. These include the cell cycle pathway, the DNA replication pathway (E10 vs. E23 and E10 vs. E28), histidine metabolism, and glycerolipid metabolism (E10 vs. E28), which are widely known to be crucial for the biological process, genetic information processing, lipid metabolism, and cellular processes. DNA replication, one of the most crucial processes in cell division [37], is also crucial for maintaining the proliferation of several skin cells, including melanocyte proliferation. Furthermore, since the up-regulated DGEs were considerably enriched in DNA replication and cell cycle pathways, it is possible that some of these DEGs are located in the nucleus and participate in cell cycle processes that increase DNA helicase activity in feather follicles and thus promote cell proliferation [38]. By performing a comprehensive analysis, we were able to confirm that the cell cycle, which includes DNA replication, was a crucial biological process in the development of the skin, particularly at the late stage, suggesting that this pathway may be connected to significant changes in the biological characteristics of the skin during its maturation. The KEGG analysis results showed that ECM–receptor interaction signaling and CAMs pathways were enriched, which was in line with the findings in the chicken and goose embryonic skins [17,19,31]. Several studies demonstrated that hair and feather follicle growth depends on the interaction between CAMs and ECM-receptors and that these pathways play a role in the interaction between the epidermis and dermis [39]. Further, the ECM–receptor interaction performs an essential function in the morphogenesis of tissues and organs, and their most important features are the upkeep of organ format and functional homeostasis, while additionally regulating several biological processes and gene expression, which include cell adhesion, proliferation, differentiation, migration, cell–cell interactions, and intracellular signaling events, which can also be related to the initiation and the formation of skin parts, which includes hair and feathers [39]. Research on Cashmere goats revealed that excessive expression of ECM and cell surface proteins was necessary for the fast growth of hair follicles through the anagen phase [28]. Moreover, we noticed that different sets of signaling pathways, such as histidine metabolism and glycerolipid metabolism and signaling, were enriched. These signaling pathways control lipid and protein synthesis and metabolism, and they have been reported in several studies related to hair and feather follicles [25,40]. Similarly, Chen et al. (2017) [40] demonstrated that besides the signal pathways that are mostly reported, other signal pathways that control lipid synthesis and metabolism, including the phosphatidylinositol signaling system, glycerolipid metabolism, O-glycan biosynthesis, inositol phosphate metabolism, glycerophospholipid metabolism, and biosynthesis of unsaturated fatty acids, occurred during the development of feather follicles in ducks. However, it has been reported that a layer of fat and oil is typically present on the surface of waterfowl feathers, and a specific amount of fat and oil is required to preserve the water-repellent characteristics of these feathers [25]. On the other hand, histidine was verified to be present in relatively significant concentrations in chicken feathers, as well as in the embryonic epidermis and feathers [41]. Therefore, we suggest that candidate genes that regulate feather development and quality could be those implicated in signal pathways for lipid and protein synthesis and metabolism.

In addition, the most important secreted genes during feather follicle initiation and development are members of the inhibitors of Wnt signaling, including WIF1 and their receptors, including Frizzled 6 (FZD6), which were also detected as significantly regulated genes in the DEGs dataset. Several studies have argued that many of the DEGs that contribute to the Wnt signal pathway are present in the dorsal tissues of geese from various breeds and at different embryonic days [15,16,17]. The Wnt signaling is known to play a crucial role in regulating follicle formation in the developing avian skin [42], and is the major pathway that regulates the patterning of skin and controls how adult and embryonic stem cells decide which cell lineages to adopt for the skin and its appendages, in addition to regulating differentiated skin cells activities [43]. Similarly, a study by Lin et al. (2006) [5] demonstrated that this pathway controls the growth of the dermis, feather bundles, and buds. Moreover, dermal Wnt signal activation and transmission paternally promote placode formation, and Wnt, with its receptors, is the first signal involved in beginning the feather follicle development program and controlling the cycle of feather follicles [44]. We noticed that the non-canonical Wnt signaling pathway’s membrane receptor protein, FZD6, was upregulated. Similar results were reported by Gong et al. (2018) [31] who discovered that the expression of FZD6 increased in chicken embryos during intra-bud morphogenesis. Furthermore, FZD6 is the mediator of the Wnt signaling pathway, and in mouse embryos, its expression in the epidermis is necessary from E11.5 to E12.5. The expression of FZD6 in the epidermis is also required for hair follicle growth [45]. Surprisingly, WIF1, a gene that inhibits the Wnt signaling pathway, was also expressed, which suggests that WIF1 might play a major part in the Wnt signaling pathway’s antagonism.

In conclusion, the enriched genes associated with cell adhesion, CAMs, the ECM receptor, and Wnt signaling pathways in KEGG and GO analyses suggest that the cells in the goose embryonic back skin are controlled to communicate, proliferate, and differentiate to shape the individual skin compartments from the homogeneous skin layers at the same time, as shown when primary feather follicles were induced in chicken skin or primary wool follicles were induced in sheep skin [14,27,31].

In addition to the signals, genes could be regulated by non-coding RNAs, including miRNAs. MiRNAs are important post-transcriptional factors and transcriptional gene network regulators that play various roles in animal epidermal morphology and hair/feather follicle growth and development and also play a crucial function in the expression of genes in several hair follicle cells lines [28,46]. Moreover, miRNAs are crucial for the development of follicles in domesticated animals such as ducks, chickens, goats, and sheep [20,30,47]. Subsequently, miRNA profiling examination of skin and feather follicles of ducks was performed by Zhang et al. (2013) [48], and they discovered that miRNAs play a key role in the function and evolution of skin. In addition, recent research suggests that controlled miRNA expression is necessary for the normal growth of hair/feather follicles. The regulation of skin and feather follicle growth is significantly influenced by the miRNA/mRNA regulatory relationship. This study examined miRNA–mRNA interaction pairings to identify the major factors involved in feather creation. Based on the integrative regulatory network, a series of DEGs and miRNAs attracted our attention, including CTGF. However, CTGF, also known as CCN2, is a crucial signaling and regulatory molecule that participates in numerous biological processes, including cell growth, angiogenesis, and wound healing [49]. In our study, the localization of CTGF by in situ hybridization showed that this gene was expressed in the skin epidermis and on the surface of the skin at the early stages of development (E10 and E13). This shows that CTGF is crucial for the development of the epidermis and dermis. Similarly, several studies reported that in the development of the epidermis and dermis in goose embryonic skin as well as the feather bud, CTGF is known to be involved [14]. Numerous studies have also shown that CTGF functions as a BMP antagonist and promotes TGF signaling during the development of hair placodes [50]. Moreover, studies in chicks have suggested that CTGF may possibly be involved in the condensation of dermal cells during placode development because it was initially discovered as a chemotactic factor for fibroblasts [14]. At the skin level, a study conducted by Mou et al. (2006) [51] on mice reported that Edar is claimed to undergo local autoregulation and signal amplification, activate CTGF, and indirectly up-regulate BMP expression in the dermis. In addition, this indicates that to avoid BMP auto stimulation, which results in an increase in follicle number and density by the end of the primary wave of follicle production, CTGF is the main BMP inhibitor employed by primary hair follicles. Our findings on the location of CTGF and its trend expression encourage these results, in which the growth of goose skin and feather follicles may also be significantly influenced by CTGF.

In our study, we identified PTCH1 as a hub gene that was highly expressed among the developmental stages and was located in skin and feather follicles during embryonic development. The PTCH1 gene is well-known for its regulatory role in embryogenesis, tissue patterning, and cell-fate decisions, acting through the Shh signaling pathway [52]. This pathway is one of the fundamental signaling pathways that help with hair follicle development and follicular bulge stem cell maintenance, as well as epidermal development, homeostasis, and repair, and PTCH1 is a marker for the SHH pathway [53]. The hedgehog ligand called Shh binds to the PTCH protein on the surface of cells, and in the absence of SHH, PTCH1 inhibits the activity of a smoothened seven-membrane receptor (SMO). Shh reduces SMO inhibition when it binds to PTCH1, which increases the amount of glioma-associated oncogene homolog (GLI) [53]. GLI target genes, including PTCH1 and GLIs, are transcribed as a result of GLI transcriptional factors, such as GLI1 and GLI2, being cleaved and transported to the nucleus [54]. Shh has been discovered to have a role in every stage of feather bud formation, including bud induction, topological shaping, of the feather filament, and feather type determination [54], and the activation of feather bud development and PTCH1 can control Shh expression. It has also been reported that in the early stages of chicken embryonic development, Shh is expressed, allowing the feather buds of the epidermis to form [55]. Moreover, Abe et al. (2017) [56] reported that PTCH1 is expressed in the epithelial and dermal sections of the early hair follicle at the start of the organogenesis phase of mouse hair follicle development, though a rise in PTCH1 expression is seen in the follicular dermal mesenchyme. In addition, Yuan et al. (2020) [57] showed that among critical module genes involved in skin and feather follicle development in duck embryos, PTCH2, PTCH1, and Shh were identified as hub genes, were significantly expressed at E15, and are enriched in the Hedgehog (HH) signaling pathway. These findings suggest that this gene is highly related to skin and feather follicle formation and is acting through the regulation of the Shh signaling pathway.

In our interaction network, we noted that a high number of small RNAs interact with the transcription factor FOXO3. In large transcriptional networks, Forkhead box, class O (FoxO) transcription factors function as signaling integrators and mediate many crucial biological processes, including DNA repair and embryonic development, and have also been linked to metabolism, cell cycle control, and apoptosis [58]. According to previous studies, the Forkhead box transcription factor class-O member, FOXO3, is a part of the transcriptional networks in the skin that regulate epidermal development. However, this gene has not been extensively studied in feather follicles, and its function in the development of feather follicles is still unknown. In the current study, the pattern of FOXO3 gene expression and its localization, as well as the growth of skin and feather follicles, were similar. It has been found that FOXO3 is generally expressed in numerous skin cell types, such as epidermal keratinocytes and dermal fibroblasts. In addition, the dermis provides nutrition to the epidermis, and both compartments work together to create skin appendages such as feather/hair follicles and sebaceous glands [59]. Furthermore, it has been demonstrated that FOXO3 functions as a co-activator of the p53 and Notch signaling pathways, respectively [60]. Keratinocyte proliferation, differentiation, and maturation must be coordinated by Notch, and Notch signaling in keratinocytes inhibits proliferation and encourages differentiation [60], and it has been extensively reported that this pathway is deeply implicated in the development of hair and feathers, so we suggest that FOXO3 may also be implicated in the production of skin and feather follicles through the activation of these pathways. Moreover, FOXO3 was detected in several studies in melanocytes, suggesting that it plays another important role in melanin biosynthesis and pigmentation [61]. In summary, the growth of skin and feather follicles may be increased by FOXO3. However, in situ hybridization assay results confirmed that FOXO3 and miR-144-y have an antagonistic targeting relationship and miR-144-y negatively regulates FOXO3. Therefore, the importance of FOXO3 in the growth of feather follicles cannot be underestimated. Recent research has revealed that several miRNAs regulate and bind to FOXO3, frequently in multiple functions [62].

The targeting effect between miRNA and its predicted target gene can be demonstrated when the expression of the miRNA is inversely correlated with the expression of the target gene, although this approach has a significant number of false positives. Therefore, we examined the regulatory impact of miR-144-y on the target gene FOXO3 and determined whether it controls FOXO3 expression using the Dual-Luciferase Reporter Gene System. The results showed that miR-144-y directly down-regulated the expression of FOXO3. However, the Dual-Luciferase Reporter assay does not reveal whether miR-144-y inhibits FOXO3 expression at the mRNA level or at the post-transcriptional stage, it only shows that the two are bound to one another. Generally, this offers a theoretical foundation for further investigation into how miR-144-y and FOXO3 control the growth of skin and feather follicles. To precisely understand how this miRNA contributes to the stimulation of goose feather follicles, additional studies are required.

## 5. Conclusions

In conclusion, in this research, we identified the major regulatory genes (FOXO3, CTGF, and PTCH1, among others) and miRNAs (miR-144-y) that could control the formation of feather follicles, and their expression pattern and localization confirmed their contribution to skin and feather follicle development. We also confirm that miR-144-y was expressed in different goose skin and feather follicles and that one of miR-144-y’s target genes is FOXO3, which verifies that the FOXO3/miR-144-y interaction played a pivotal role in controlling skin and feather follicle development. To summarize, our findings offer a new understanding of the molecular regulation of skin and feather follicle development processes in Zhedong white goose.

## Figures and Tables

**Figure 1 animals-12-02099-f001:**
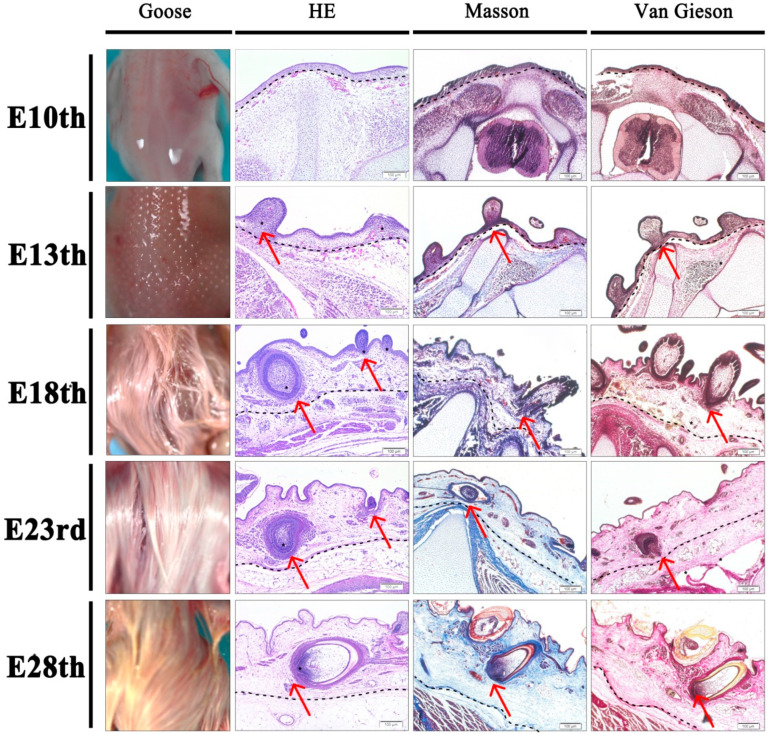
Developmental pattern of skin and feather follicles in embryonic Zhedong white geese. The vertical rows show the time gradient, and the horizontal rows show the results of different stains. Magnified: 100×; Bar: 100 μm.

**Figure 2 animals-12-02099-f002:**
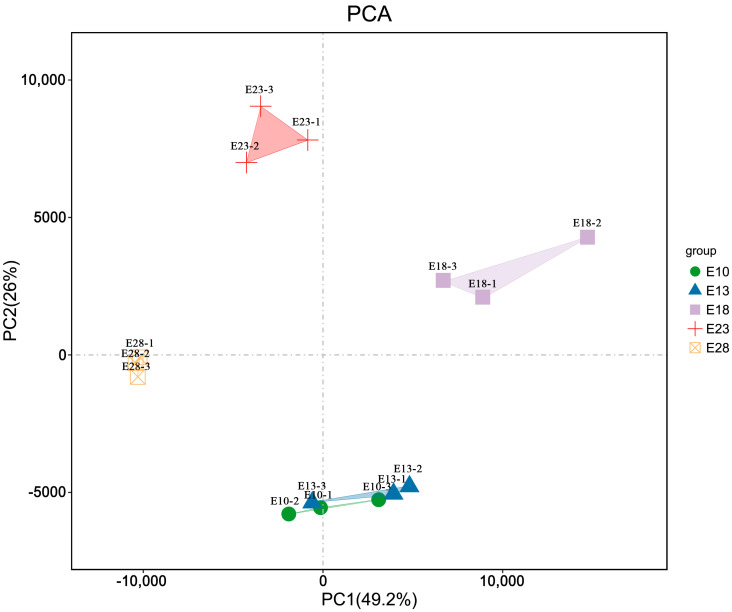
Principal Component Analysis (PCA) relationships between groups. The PC1 coordinate represents the first principal component and the percentage in brackets shows the contribution of the first principal component to the variation in the samples; the PC2 coordinate indicates the second principal component and the percentage in brackets indicates the contribution of the second principal component to the variation in the samples. The colored points in the graph indicate the individual samples.

**Figure 3 animals-12-02099-f003:**
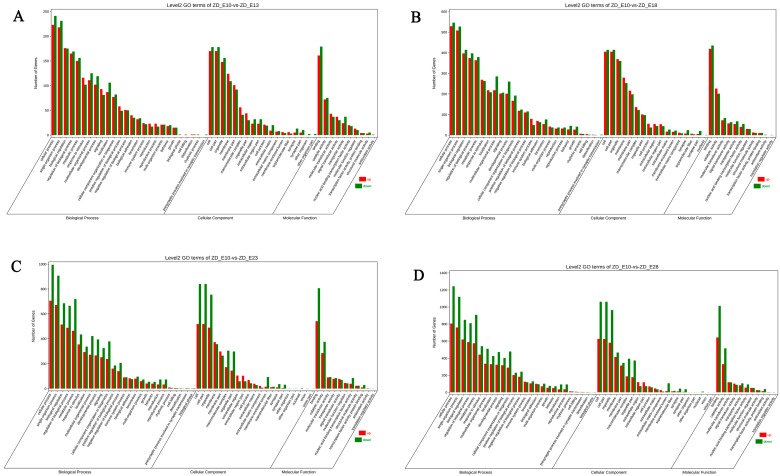
GO analysis of genetic differences in skin and feather follicle development in Zhedong white geese. (**A**–**D**) shows the comparison group of E13, E18, E23, and E28 with E10. The results are summarized into three major groups: Biological process, cellular component, and molecular function. The *Y*-axis displays the percentage of genes, while the *X*-axis represents the second level term of the gene ontology.

**Figure 4 animals-12-02099-f004:**
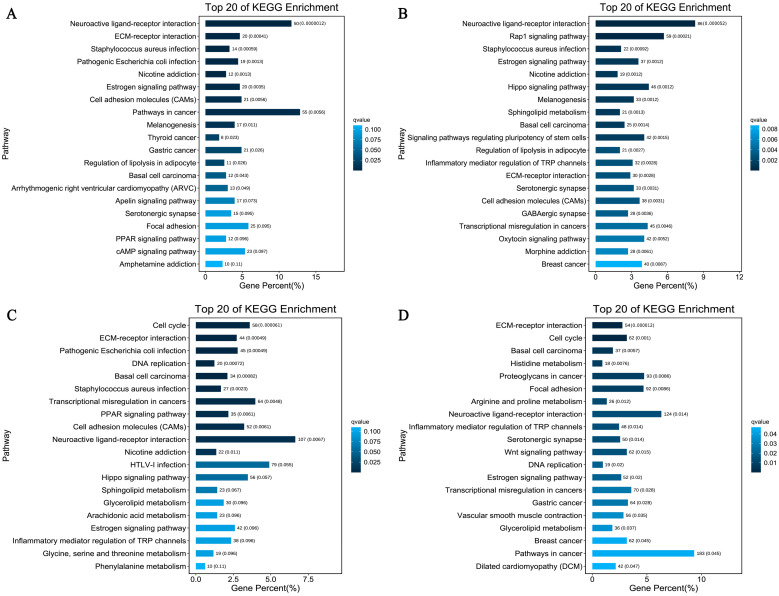
KEGG analysis of genetic differences in skin and feather follicle development in Zhedong white geese. (**A**–**D**) shows the comparison group of E13, E18, E23, and E28 with E10. The abscissa represents the number of enriched genes, and the ordinates represent signal pathways.

**Figure 5 animals-12-02099-f005:**
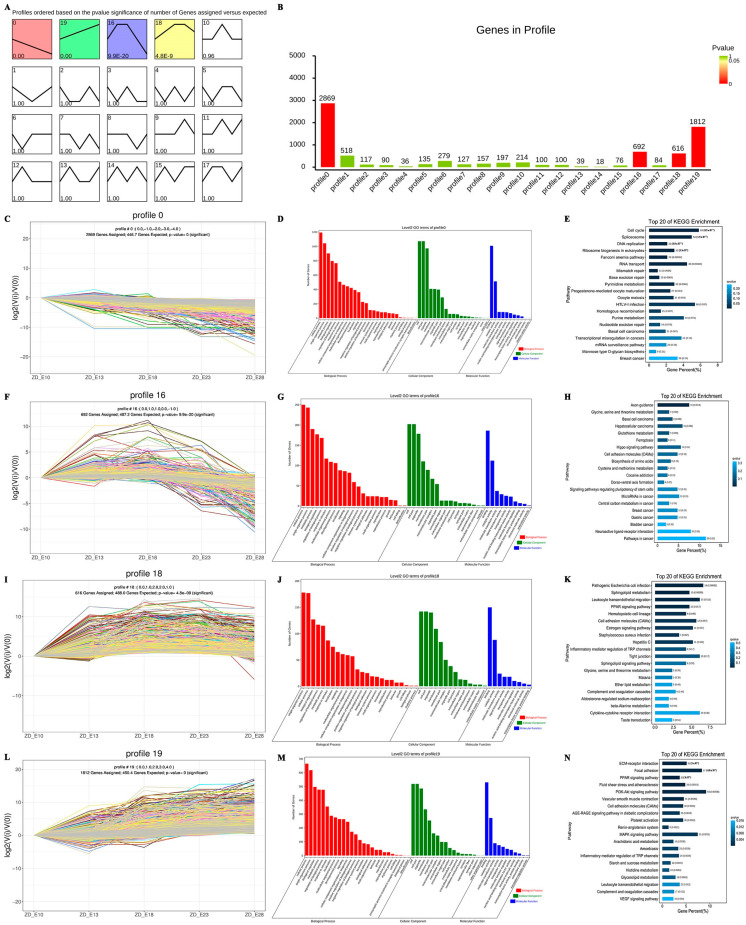
Trend analysis of differential genes for skin and feather follicle development in Zhedong white geese. (**A**) Summary of trends in differentially expressed genes (DEGs). The colored profiles (*p* < 0.05) indicate significant enrichment and, conversely, non-significant enrichment. Similar color patterns are shown in profiles with the same expressive tendency. (**B**) The number of genes in a gene profile. (**C**–**E**) Trend plots, GO and KEGG analysis of profile 0. (**F**–**H**) Trend plots, GO and KEGG analysis of profile 16. (**I**–**K**) Trend plots, GO and KEGG analysis of profile 18. (**L**–**N**) Trend plots, GO and KEGG analysis of profile 19. Profiles C-F-E-L: Each *X*-axis indicates the embryonic stages (E10, E13, E18, E23, and E28); the *Y*-axis displays variations in expression. Four profiles (profiles C-F-E-L) represent the major transcriptional trajectories.

**Figure 6 animals-12-02099-f006:**
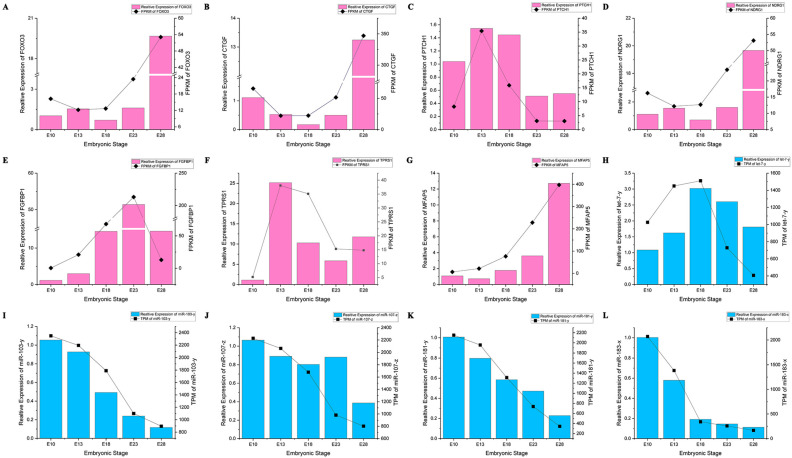
Validation of the RNA-Seq and miRNA-Seq data by RT-qPCR using 7 selected DEGs and 5 selected miRNAs. (**A**–**G**) The relative expression pattern and FPKM of seven genes (FOXO3, CTGF, PTCH1, NDRG1, FGFBP1, TPRS1, and MFAP5). (**H**–**L**) The relative expression pattern and TPM of five miRNAs (let-7-y, miR-103-y, miR-107-z, miR-181-y, and miR-183-x). As reference genes for miRNA and mRNA testing, respectively, U6, 18S rRNA, and GAPDH were used. The results are shown as the means ± SEM of three replicates.

**Figure 7 animals-12-02099-f007:**
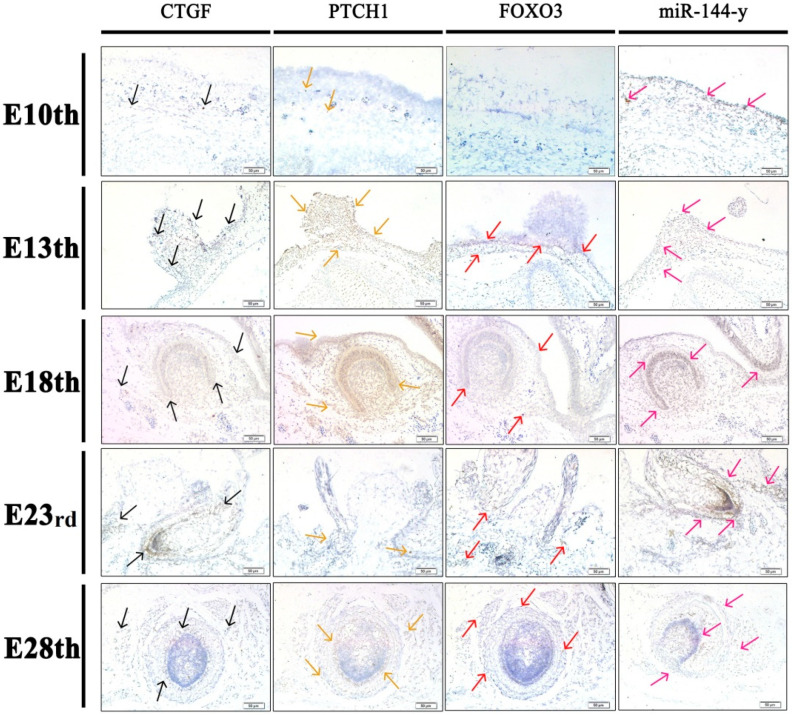
In situ hybridization of selected genes and miRNA. The horizontal column shows the mRNA/miRNA expression localization of CTGF, PTCH1, FOXO3, and miR-144-y, and the vertical column shows the expression based on different embryonic stages. Magnified: 20×; Bar: 50 μm.

**Figure 8 animals-12-02099-f008:**
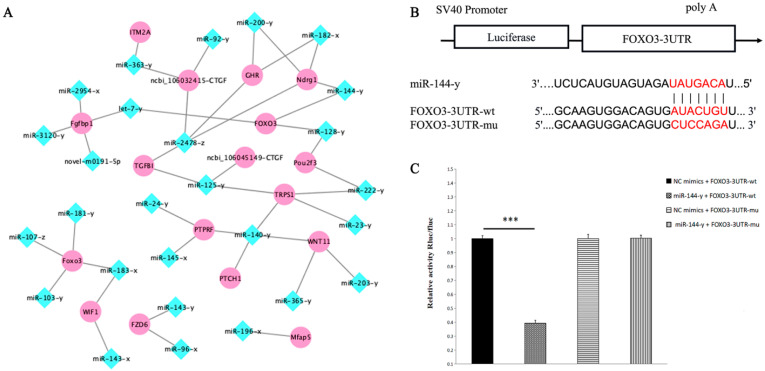
miRNA-mRNA interaction network and targeting validation. (**A**) Part of the interaction diagram between miRNA and target mRNA; the diamonds in blue correspond to the miRNAs and the circles in pink indicate the gene abbreviations. (**B**) Schematic representation of miR-144-y binding to the FOXO3-3′UTR target site. (**C**) Dual-luciferase reporter gene assay of miR-144-y interacting with FOXO3-3′UTR. *** *p* < 0.001.

**Table 1 animals-12-02099-t001:** Primer list for qRT-PCR validation.

Gene	Primer Sequence	Application
let-7-y	GTCGTATCCAGTGCAGGGTCCGAGGTATTCGCACTGGATACGACAGGAAG	RT-PCR
	F: CGCGCGCTATACAGTCTACTGT	qPCR
miR-103-y	GTCGTATCCAGTGCAGGGTCCGAGGTATTCGCACTGGATACGACTCATAG	RT-PCR
	F: GCGAGCAGCATTGTACAGGG	qPCR
miR-107-z	GTCGTATCCAGTGCAGGGTCCGAGGTATTCGCACTGGATACGACTCATAG	RT-PCR
	F: GCGAGCAGCATTGTACAGGG	qPCR
miR-181-y	GTCGTATCCAGTGCAGGGTCCGAGGTATTCGCACTGGATACGACGGTACA	RT-PCR
	F: GCGACCATCGACCGTTGAT	qPCR
miR-183-x	GTCGTATCCAGTGCAGGGTCCGAGGTATTCGCACTGGATACGACCAGTGA	RT-PCR
	F: CGCGTATGGCACTGGTAGAAT	qPCR
U6	F: CTCGCTTCGGCAGCACATATACTA R: CGAATTTGCGTGTCATCCTTGC	qPCR
18S rRNA	F: GCATGGCCGTTCTTAGTTGG R: GAACGCCACTTGTCCCTCTA	qPCR
GAPDH	F: CGTGTGGTGGACTTGATGGT R: AAGGGAACAGAACTGGCCTC	qPCR
FOXO3	F: ATCACGAAGTCTGGGGCTTG R: ACGAGGGCGAATTTTAGGCA	qPCR
CTGF	F: AGCGTGAAGACCTACAGAGC R: CATGATCTCCCCATCAGGGC	qPCR
PTCH1	F: CCTGTGCCTCAGTTTCTCGT R: TGCACTTACCTGGCACCTTT	qPCR
NDRG1	F: ACTCGCCTCCTACCAGACTT R: ACCGCAAAGTGCTGAGTGAT	qPCR
FGFBP1	F: CAGCAAGTTCTGGTGCGAGT R: GAGGATGCTTCTGGGGTCCT	qPCR
TPRS1	F: GCCTTATGAAGTCAATGCTGG R: ACATGCGTGCAAAGTTCCTC	qPCR
MFAP5	F: TGTGGCTGCATGTGATTTACC R: GTCTTCATCGTGTTGCCCTC	qPCR

The candidate genes: FOXO3: forkhead box O3; CTGF: Connective tissue growth factor; PTCH1: Protein parched homolog1; NDRG1: N-Myc down regulated Gene1; FGFBP1: Fibroblast growth factor binding protein 1; TPRS1: Transcriptional Repressor GATA Binding 1; MFAP5: Microfibril Associated Protein 5; GAPDH: Glyceraldehyde-3-phosphate dehydrogenase gene. F denotes forward primers and R denotes reverse primers.

**Table 2 animals-12-02099-t002:** Probe sequence for in situ hybridization.

Target Gene	Sequence
CTGF	TCCTTGGGCTCGTCACAGACCCACTCCTCG
PTCH1	GGTCGCAGCCCTTCCCTCACTTCCCGTTTG
FOXO3	CAGCATGGGAGAAAGCGGAGCGTCATCGTC
miR-144-y	AGAGTACATCATCTATACTGTA

## Data Availability

The raw data of RNA-seq and miRNA-seq utilized and analyzed for this study are publicly available from CNGBdb. Available online: https://db.cngb.org (accessed on 16 May 2022) with project ID CNP0003038.

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
