# Peer review of "Transcriptional Characteristics Showed That miR-144-y/FOXO3 Participates in Embryonic Skin and Feather Follicle Development in Zhedong White Goose"

_animals, 2022, doi:10.3390/ani12162099_

Round 1
Reviewer 1 Report
Comments for the manuscript: “Transcriptional Characteristics Showed that miR-144-y/FOXO3 Participates in Embryonic Skin and Feather Follicle Develop-3 ment in Zhedong White Goose” The paper has the practical utility but authors need to incorporate some minor changes as highlighted in the manuscript.

Author Response
Thank you for your valuable comments and suggestions on this article. Throughout your comments, we have modified and corrected all the highlighted minor changes in the manuscript. Thank you again for your comments.
Please see the attachment of the manuscript.

Reviewer 2 Report
In this paper, Mabrouk and colleagues, via transcriptome and next-generation sequencing (NGS) network analyses, analyzed genes involved in skin and feather follicle development of Zhedong White goose. They found several differentially expressed genes, involved in pathways (such as Wnt, cell adhesion, DNA replication) regulating feather growth and development.
Moreover, the localization of FOXO3, CTGF, PTCH1 and miR-144-y by in situ hybridization showed spatial-temporal expression patterns and that FOXO3 and miR-144-y have an antagonistic targeting relationship.
The paper is interesting, and the experimental approach appropriate. I recommend for its publication in Animals prior these minor revisions:
- In line 29, what did the author mean with “improve the genome of Zhedong White goose”?
- In line 32, the authors should specify what the “four stages” are referred to;
- In line 32 (but also elsewhere in the text), what is the meaning of “altered genes”?
- Why just males where selected for the study?
- In line 111, as the authors are reporting the hydration step, the used alcohol should be listed starting from the most concentrated to the less;
- In line 144, why the authors enriched also prokaryotic mRNA?
- In section 2.15, please add this information: the thickness of the slides, the protocol for the probe synthesis, the reference code of the anti-DIG-HRP;
- In Figure 1 symbols should be added to indicate what the authors are describing; in Figure 7, different symbols should be used to indicate different localization of the analyzed genes;
- The labels in Figure 3 are difficult to read, please improve it;
- The Discussion is too long and difficult to follow; I suggest to reduce it.
Author Response
Thank you for your valuable comments and suggestions on this article.
Please see the attachment.

Round 2
Reviewer 1 Report
May be accept in present form.
Reviewer 2 Report
The referee is satisfi ed by authors responses tha paper is now accetteble for piublication on Inms